# Decoupling "when to update" from "how to update"

**Eran Malach**
School of Computer Science
The Hebrew University, Israel
eran.malach@mail.huji.ac.il

**Shai Shalev-Shwartz**
School of Computer Science
The Hebrew University, Israel
shais@cs.huji.ac.il

## Abstract

Deep learning requires data. A useful approach to obtain data is to be creative and mine data from various sources, that were created for different purposes. Unfortunately, this approach often leads to noisy labels. In this paper, we propose a meta algorithm for tackling the noisy labels problem. The key idea is to decouple "when to update" from "how to update". We demonstrate the effectiveness of our algorithm by mining data for gender classification by combining the Labeled Faces in the Wild (LFW) face recognition dataset with a textual genderizing service, which leads to a noisy dataset. While our approach is very simple to implement, it leads to state-of-the-art results. We analyze some convergence properties of the proposed algorithm.

## 1   Introduction

In recent years, deep learning achieves state-of-the-art results in various different tasks, however, neural networks are mostly trained using supervised learning, where a massive amount of labeled data is required. While collecting unlabeled data is relatively easy given the amount of data available on the web, providing accurate labeling is usually an expensive task. In order to overcome this problem, data science becomes an art of extracting labels out of thin air. Some popular approaches to labeling are crowdsourcing, where the labeling is not done by experts, and mining available meta-data, such as text that is linked to an image in a webpage. Unfortunately, this gives rise to a problem of abundant noisy labels - labels may often be corrupted [19], which might deteriorate the performance of neural-networks [12].

Let us start with an intuitive explanation as to why noisy labels are problematic. Common neural network optimization algorithms start with a random guess of what the classifier should be, and then iteratively update the classifier based on stochastically sampled examples from a given dataset, optimizing a given loss function such as the hinge loss or the logistic loss. In this process, wrong predictions lead to an update of the classifier that would hopefully result in better classification performance. While at the beginning of the training process the predictions are likely to be wrong, as the classifier improves it will fail on less and less examples, thus making fewer and fewer updates. On the other hand, in the presence of noisy labels, as the classifier improves the effect of the noise increases - the classifier may give correct predictions, but will still have to update due to wrong labeling. Thus, in an advanced stage of the training process the majority of the updates may actually be due to wrongly labeled examples, and therefore will not allow the classifier to further improve.

To tackle this problem, we propose to decouple the decision of "when to update" from the decision of "how to update". As mentioned before, in the presence of noisy labels, if we update only when the classifier's prediction differs from the available label, then at the end of the optimization process, these few updates will probably be mainly due to noisy labels. We would therefore like a different update criterion, that would let us decide whether it is worthy to update the classifier based on a given example. We would like to preserve the behavior of performing many updates at the beginning of the training process but only a few updates when we approach convergence. To do so, we

suggest to train two predictors, and perform update steps only in case of disagreement between them. This way, when the predictors get better, the "area" of their disagreement gets smaller, and updates are performed only on examples that lie in the disagreement area, therefore preserving the desired behavior of the standard optimization process. On the other hand, since we do not perform an update based on disagreement with the label (which may be due to a problem in the label rather than a problem in the predictor), this method keeps the effective amount of noisy labels seen throughout the training process at a constant rate.

The idea of deciding "when to update" based on a disagreement between classifiers is closely related to approaches for active learning and selective sampling - a setup in which the learner does not have unlimited access to labeled examples, but rather has to query for each instance's label, provided at a given cost (see for example [34]). Specifically, the well known query-by-committee algorithm maintains a version space of hypotheses and at each iteration, decides whether to query the label of a given instance by sampling two hypotheses uniformly at random from the version space [35, 14]. Naturally, maintaining the version space of deep networks seems to be intractable. Our algorithm maintains only two deep networks. The difference between them stems from the random initialization. Therefore, unlike the original query-by-committee algorithm, that samples from the version space at every iteration, we sample from the original hypotheses class only once (at the initialization), and from there on, we update these two hypotheses using the backpropagation rule, when they disagree on the label. To the best of our knowledge, this algorithm was not proposed/analyzed previously, not in the active learning literature and especially not as a method for dealing with noisy labels.

To show that this method indeed improves the robustness of deep learning to noisy labels, we conduct an experiment that aims to study a real-world scenario of acquiring noisy labels for a given dataset. We consider the task of gender classification based on images. We did not have a dedicated dataset for this task. Instead, we relied on the Labeled Faces in the Wild (LFW) dataset, which contains images of different people along with their names, but with no information about their gender. To find the gender for each image, we use an online service to match a gender to a given name (as is suggested by [25]), a method which is naturally prone to noisy labels (due to unisex names). Applying our algorithm to an existing neural network architecture reduces the effect of the noisy labels, achieving better results than similar available approaches, when tested on a clean subset of the data. We also performed a controlled experiment, in which the base algorithm is the perceptron, and show that using our approach leads to a noise resilient algorithm, which can handle an extremely high label noise rates of up to $40\%$. The controlled experiments are detailed in Appendix B.

In order to provide theoretical guarantees for our meta algorithm, we need to tackle two questions: 1. does this algorithm converge? and if so, how quickly? and 2. does it converge to an optimum? We give a positive answer to the first question, when the base algorithm is the perceptron and the noise is label flip with a constant probability. Specifically, we prove that the expected number of iterations required by the resulting algorithm equals (up to a constant factor) to that of the perceptron in the noise-free setting. As for the second question, clearly, the convergence depends on the initialization of the two predictors. For example, if we initialize the two predictors to be the same predictor, the algorithm will not perform any updates. Furthermore, we derive lower bounds on the quality of the solution even if we initialize the two predictors at random. In particular, we show that for some distributions, the algorithm's error will be bounded away from zero, even in the case of linearly separable data. This raises the question of whether a better initialization procedure may be helpful. Indeed, we show that for the same distribution mentioned above, even if we add random label noise, if we initialize the predictors by performing few vanilla perceptron iterations, then the algorithm performs much better. Despite this worst case pessimism, we show that empirically, when working with natural data, the algorithm converges to a good solution. We leave a formal investigation of distribution dependent upper bounds to future work.

## 2   Related Work

The effects of noisy labels was vastly studied in many different learning algorithms (see for example the survey in [13]), and various solutions to this problem have been proposed, some of them with theoretically provable bounds, including methods like statistical queries, boosting, bagging and more [21, 26, 7, 8, 29, 31, 23, 27, 3]. Our focus in this paper is on the problem of noisy labels in the context of deep learning. Recently, there have been several works aiming at improving the resilience of deep

learning to noisy labels. To the best of our knowledge, there are four main approaches. The first changes the loss function. The second adds a layer that tries to mimic the noise behavior. The third groups examples into buckets. The fourth tries to clean the data as a preprocessing step. Beyond these approaches, there are methods that assume a small clean data set and another large, noisy, or even unlabeled, data set [30, 6, 38, 1]. We now list some specific algorithms from these families.

[33] proposed to change the cross entropy loss function by adding a regularization term that takes into account the current prediction of the network. This method is inspired by a technique called minimum entropy regularization, detailed in [17, 16]. It was also found to be effective by [12], which suggested a further improvement of this method by effectively increasing the weight of the regularization term during the training procedure.

[28] suggested to use a probabilistic model that models the conditional probability of seeing a wrong label, where the correct label is a latent variable of the model. While [28] assume that the probability of label-flips between classes is known in advance, a follow-up work by [36] extends this method to a case were these probabilities are unknown. An improved method, that takes into account the fact that some instances might be more likely to have a wrong label, has been proposed recently in [15]. In particular, they add another softmax layer to the network, that can use the output of the last hidden layer of the network in order to predict the probability of the label being flipped. Unfortunately, their method involves optimizing the biases of the additional softmax layer by first training it on a simpler setup (without using the last hidden layer), which implies two-phase training that further complicates the optimization process. It is worth noting that there are some other works that suggest methods that are very similar to [36, 15], with a slightly different objective or training method [5, 20], or otherwise suggest a complicated process which involves estimation of the class-dependent noise probabilities [32]. Another method from the same family is the one described in [37], who suggests to differentiate between "confusing" noise, where some features of the example make it hard to label, or otherwise a completely random label noise, where the mislabeling has no clear reason.

[39] suggested to train the network to predict labels on a randomly selected group of images from the same class, instead of classifying each image individually. In their method, a group of images is fed as an input to the network, which merges their inner representation in a deeper level of the network, along with an attention model added to each image, and producing a single prediction. Therefore, noisy labels may appear in groups with correctly labeled examples, thus diminishing their impact. The final setup is rather complicated, involving many hyper-parameters, rather than providing a simple plug-and-play solution to make an existing architecture robust to noisy labels.

From the family of preprocessing methods, we mention [4, 10], that try to eliminate instances that are suspected to be mislabeled. Our method shares a similar motivation of disregarding contaminated instances, but without the cost of complicating the training process by a preprocessing phase.

In our experiment we test the performance of our method against methods that are as simple as training a vanilla version of neural network. In particular, from the family of modified loss function we chose the two variants of the regularized cross entropy loss suggested by [33] (soft and hard bootsrapping). From the family of adding a layer that models the noise, we chose to compare to one of the models suggested in [15] (which is very similar to the model proposed by [36]), because this model does not require any assumptions or complication of the training process. We find that our method outperformed all of these competing methods, while being extremely simple to implement.

Finally, as mentioned before, our "when to update" rule is closely related to approaches for active learning and selective sampling, and in particular to the query-by-committee algorithm. In [14] a thorough analysis is provided for various base algorithms implementing the query-by-committee update rule, and particularly they analyze the perceptron base algorithm under some strong distributional assumptions. In other works, an ensemble of neural networks is trained in an active learning setup to improve the generalization of neural networks [11, 2, 22]. Our method could be seen as a simplified member of ensemble methods. As mentioned before, our motivation is very different than the active learning scenario, since our main goal is dealing with noisy labels, rather than trying to reduce the number of label queries. To the best of our knowledge, the algorithm we propose was not used or analyzed in the past for the purpose of dealing with noisy labels in deep learning.

# 3   METHOD

As mentioned before, to tackle the problem of noisy labels, we suggest to change the update rule commonly used in deep learning optimization algorithms in order to decouple the decision of "when to update" from "how to update". In our approach, the decision of "when to update" does not depend on the label. Instead, it depends on a disagreement between two different networks. This method could be generally thought of as a meta-algorithm that uses two base classifiers, performing updates according to a base learning algorithm, but only on examples for which there is a disagreement between the two classifiers.

To put this formally, let $\mathcal{X}$ be an instance space and $\mathcal{Y}$ be the label space, and assume we sample examples from a distribution $\tilde{\mathcal{D}}$ over $\mathcal{X} \times \mathcal{Y}$, with possibly noisy labels. We wish to train a classifier $h$, coming from a hypothesis class $\mathcal{H}$. We rely on an update rule, $U$, that updates $h$ based on its current value as well as a mini-batch of $b$ examples. The meta algorithm receives as input a pair of two classifiers, $h_1, h_2 \in \mathcal{H}$, the update rule, $U$, and a mini batch size, $b$. A pseudo-code is given in Algorithm 1.

Note that we do not specify how to initialize the two base classifiers, $h_1, h_2$. When using deep learning as the base algorithm, the easiest approach is maybe to perform a random initialization. Another approach is to first train the two classifiers while following the regular "when to update" rule (which is based on the label $y$), possibly training each classifier on a different subset of the data, and switching to the suggested update rule only in an advanced stage of the training process. We later show that the second approach is preferable.

At the end of the optimization process, we can simply return one of the trained classifiers. If a small accurately labeled test data is available, we can choose to return the classifier with the better accuracy on the clean test data.

---

**Algorithm 1** Update by Disagreement

> **input**:
>     an update rule $U$
>     batch size $b$
>     two initial predictors $h_1, h_2 \in \mathcal{H}$
> **for** $t = 1, 2, \ldots, N$ **do**
>     draw mini-batch $(x_1, y_1), \ldots, (x_b, y_b) \sim \tilde{\mathcal{D}}^b$
>     let $S = \{(x_i, y_i) : h_1(x_i) \neq h_2(x_i)\}$
>     $h_1 \leftarrow U(h_1, S)$
>     $h_2 \leftarrow U(h_2, S)$
> **end for**

---

# 4   Theoretical analysis

Since a convergence analysis for deep learning is beyond our reach even in the noise-free setting, we focus on analyzing properties of our algorithm for linearly separable data, which is corrupted by random label noise, and while using the perceptron as a base algorithm.

Let $\mathcal{X} = \{x \in \mathbb{R}^d : \|x\| \leq 1\}$, $\mathcal{Y} = \{\pm 1\}$, and let $\mathcal{D}$ be a probability distribution over $\mathcal{X} \times \mathcal{Y}$, such that there exists $w^*$ for which $\mathcal{D}(\{(x, y) : y\langle w^*, x\rangle < 1\}) = 0$. The distribution we observe, denoted $\tilde{\mathcal{D}}$, is a noisy version of $\mathcal{D}$. Specifically, to sample $(x, \tilde{y}) \sim \tilde{\mathcal{D}}$ one should sample $(x, y) \sim \mathcal{D}$ and output $(x, y)$ with probability $1 - \mu$ and $(x, -y)$ with probability $\mu$. Here, $\mu$ is in $[0, 1/2)$.

Finally, let $\mathcal{H}$ be the class of linear classifiers, namely, $\mathcal{H} = \{x \mapsto \text{sign}(\langle w, x\rangle) : w \in \mathbb{R}^d\}$. We use the perceptron's update rule with mini-batch size of 1. That is, given the classifier $w_t \in \mathbb{R}^d$, the update on example $(x_t, y_t) \in \mathcal{X} \times \mathcal{Y}$ is: $w_{t+1} = U(w_t, (x_t, y_t)) := w_t + y_t x_t$.

As mentioned in the introduction, to provide a full theoretical analysis of this algorithm, we need to account for two questions:

1. does this algorithm converge? and if so, how quickly?
2. does it converge to an optimum?

Theorem 1 below provides a positive answer for the first question. It shows that the number of updates of our algorithm is only larger by a constant factor (that depends on the initial vectors and the amount of noise) relatively to the bound for the vanilla perceptron in the noise-less case.

**Theorem 1** *Suppose that the "Update by Disagreement" algorithm is run on a sequence of random $N$ examples from $\tilde{\mathcal{D}}$, and with initial vectors $w_0^{(1)}, w_0^{(2)}$. Denote $K = \max_i \|w_0^{(i)}\|$. Let $T$ be the number of updates performed by the "Update by Disagreement" algorithm.*
*Then, $\mathbb{E}[T] \leq \frac{3(4K+1)}{(1-2\mu)^2} \|w^*\|^2$ where the expectation is w.r.t. the randomness of sampling from $\tilde{\mathcal{D}}$.*

**Proof** It will be more convenient to rewrite the algorithm as follows. We perform $N$ iterations, where at iteration $t$ we receive $(x_t, \tilde{y}_t)$, and update $w_{t+1}^{(i)} = w_t^{(i)} + \tau_t \tilde{y}_t x_t$ , where

$$
\tau_t = \begin{cases} 1 & \text{if } \text{sign}(\langle w_t^{(1)}, x_t \rangle) \neq \text{sign}(\langle w_t^{(2)}, x_t \rangle) \\ 0 & \text{otherwise} \end{cases}
$$

Observe that we can write $\tilde{y}_t = \theta_t y_t$, where $(x_t, y_t) \sim \mathcal{D}$, and $\theta_t$ is a random variables with $\mathbb{P}[\theta_t = 1] = 1 - \mu$ and $\mathbb{P}[\theta_t = -1] = \mu$. We also use the notation $v_t = y_t \langle w^*, x_t \rangle$ and $\tilde{v}_t = \theta_t v_t$. Our goal is to upper bound $\bar{T} := \mathbb{E}[T] = \mathbb{E}[\sum_t \tau_t]$.

We start with showing that

$$
\mathbb{E}\left[ \sum_{t=1}^{N} \tau_t \tilde{v}_t \right] \geq (1 - 2\mu)\overline{T} \tag{1}
$$

Indeed, since $\theta_t$ is independent of $\tau_t$ and $v_t$, we get that:

$$
\mathbb{E}[\tau_t \tilde{v}_t] = \mathbb{E}[\tau_t \theta_t v_t] = \mathbb{E}[\theta_t] \cdot \mathbb{E}[\tau_t v_t] = (1 - 2\mu)\,\mathbb{E}[\tau_t v_t] \geq (1 - 2\mu)\,\mathbb{E}[\tau_t]
$$

where in the last inequality we used the fact that $v_t \geq 1$ with probability 1 and $\tau_t$ is non-negative. Summing over $t$ we obtain that Equation 1 holds.

Next, we show that for $i \in \{1, 2\}$,

$$
\|w_t^{(i)}\|^2 \leq \|w_0^{(i)}\|^2 + \sum_{t=1}^{N} \tau_t (2\|w_0^{(2)} - w_0^{(1)}\| + 1) \tag{2}
$$

Indeed, since the update of $w_{t+1}^{(1)}$ and $w_{t+1}^{(2)}$ is identical, we have that $\|w_{t+1}^{(1)} - w_{t+1}^{(2)}\| = \|w_0^{(1)} - w_0^{(2)}\|$ for every $t$. Now, whenever $\tau_t = 1$ we have that either $y_t \langle w_{t-1}^{(1)}, x_t \rangle \leq 0$ or $y_t \langle w_{t-1}^{(2)}, x_t \rangle \leq 0$. Assume w.l.o.g. that $y_t \langle w_{t-1}^{(1)}, x_t \rangle \leq 0$. Then,

$$
\|w_t^{(1)}\|^2 = \|w_{t-1}^{(1)} + y_t x_t\|^2 = \|w_{t-1}^{(1)}\|^2 + 2 y_t \langle w_{t-1}^{(1)}, x_t \rangle + \|x_t\|^2 \leq \|w_{t-1}^{(1)}\|^2 + 1
$$

Second,

$$
\begin{aligned}
\|w_t^{(2)}\|^2 &= \|w_{t-1}^{(2)} + y_t x_t\|^2 = \|w_{t-1}^{(2)}\|^2 + 2 y_t \langle w_{t-1}^{(2)}, x_t \rangle + \|x_t\|^2 \\
&\leq \|w_{t-1}^{(2)}\|^2 + 2 y_t \langle w_{t-1}^{(2)} - w_{t-1}^{(1)}, x_t \rangle + \|x_t\|^2 \\
&\leq \|w_{t-1}^{(2)}\|^2 + 2\|w_{t-1}^{(2)} - w_{t-1}^{(1)}\| + 1 = \|w_{t-1}^{(2)}\|^2 + 2\|w_0^{(2)} - w_0^{(1)}\| + 1
\end{aligned}
$$

Therefore, the above two equations imply $\forall i \in \{1, 2\}$, $\|w_t^{(i)}\|^2 \leq \|w_{t-1}^{(i)}\|^2 + 2\|w_0^{(2)} - w_0^{(1)}\| + 1$. Summing over $t$ we obtain that Equation 2 holds.

Equipped with Equation 1 and Equation 2 we are ready to prove the theorem.
Denote $K = \max_i \|w_0^{(i)}\|$ and note that $\|w_0^{(2)} - w_0^{(1)}\| \leq 2K$. We prove the theorem by providing upper and lower bounds on $\mathbb{E}[\langle w_t^{(i)}, w^* \rangle]$. Combining the update rule with Equation 1 we get:

$$
\mathbb{E}[\langle w_t^{(i)}, w^* \rangle] = \langle w_0^{(i)}, w^* \rangle + \mathbb{E}\left[ \sum_{t=1}^{N} \tau_t \tilde{v}_t \right] \geq \langle w_0^{(i)}, w^* \rangle + (1 - 2\mu)\bar{T} \geq -K\|w^*\| + (1 - 2\mu)\bar{T}
$$

To construct an upper bound, first note that Equation 2 implies that

$$
\mathbb{E}[\|w_t^{(i)}\|^2] \leq \|w_0^{(i)}\|^2 + (2\|w_0^{(2)} - w_0^{(1)}\| + 1)\bar{T} \leq K^2 + (4K + 1)\bar{T}
$$

Using the above and Jensen's inequality, we get that

$$\mathbb{E}[\langle w_t^{(i)}, w^* \rangle] \leq \mathbb{E}[\|w_t^{(i)}\| \|w^*\|] \leq \|w^*\| \sqrt{\mathbb{E}[\|w_t^{(i)}\|^2]} \leq \|w^*\| \sqrt{K^2 + (4K+1)\bar{T}}$$

Comparing the upper and lower bounds, we obtain that

$$-K\|w^*\| + (1 - 2\mu)\bar{T} \leq \|w^*\| \sqrt{K^2 + (4K+1)\bar{T}}$$

Using $\sqrt{a+b} \leq \sqrt{a} + \sqrt{b}$, the above implies that

$$(1 - 2\mu)\bar{T} - \|w^*\| \sqrt{(4K+1)} \sqrt{\bar{T}} - 2K\|w^*\| \leq 0$$

Denote $\alpha = \|w^*\| \sqrt{(4K+1)}$, then the above also implies that $(1-2\mu)\bar{T} - \alpha\sqrt{\bar{T}} - \alpha \leq 0$.
Denote $\beta = \alpha/(1-2\mu)$, using standard algebraic manipulations, the above implies that

$$\bar{T} \leq \beta + \beta^2 + \beta^{1.5} \leq 3\beta^2,$$

where we used the fact that $\|w^*\|$ must be at least 1 for the separability assumption to hold, hence $\beta \geq 1$. This concludes our proof. ∎

The above theorem tells us that our algorithm converges quickly. We next address the second question, regarding the quality of the point to which the algorithm converges. As mentioned in the introduction, the convergence must depend on the initial predictors. Indeed, if $w_0^{(1)} = w_0^{(2)}$, then the algorithm will not make any updates. The next question is what happens if we initialize $w_0^{(1)}$ and $w_0^{(2)}$ at random. The lemma below shows that this does not suffice to ensure convergence to the optimum, even if the data is linearly separable without noise. The proof for this lemma is given in Appendix A.

**Lemma 1** *Fix some $\delta \in (0,1)$ and let $d$ be an integer greater than $40 \log(1/\delta)$. There exists a distribution over $\mathbb{R}^d \times \{\pm 1\}$, which is separable by a weight vector $w^*$ for which $\|w^*\|^2 = d$, such that running the "Update by Disagreement" algorithm, with the perceptron as the underlying update rule, and with every coordinate of $w_0^{(1)}, w_0^{(2)}$ initialized according to any symmetric distribution over $\mathbb{R}$, will yield a solution whose error is at least $1/8$, with probability of at least $1 - \delta$.*

Trying to circumvent the lower bound given in the above lemma, one may wonder what would happen if we will initialize $w_0^{(1)}, w_0^{(2)}$ differently. Intuitively, maybe noisy labels are not such a big problem at the beginning of the learning process. Therefore, we can initialize $w_0^{(1)}, w_0^{(2)}$ by running the vanilla perceptron for several iterations, and only then switch to our algorithm. Trivially, for the distribution we constructed in the proof of Lemma 1, this approach will work just because in the noise-free setting, both $w_0^{(1)}$ and $w_0^{(2)}$ will converge to vectors that give the same predictions as $w^*$. But, what would happen in the noisy setting, when we flip the label of every example with probability of $\mu$? The lemma below shows that the error of the resulting solution is likely to be order of $\mu^3$. Here again, the proof is given in Appendix A.

**Lemma 2** *Consider a vector $w^* \in \{\pm 1\}^d$ and the distribution $\tilde{\mathcal{D}}$ over $\mathbb{R}^d \times \{\pm 1\}$ such that to sample a pair $(x, \tilde{y})$ we first choose $x$ uniformly at random from $\{e_1, \ldots, e_d\}$, set $y = \langle w^*, e_i \rangle$, and set $\tilde{y} = y$ with probability $1 - \mu$ and $\tilde{y} = -y$ with probability $\mu$. Let $w_0^{(1)}, w_0^{(2)}$ be the result of running the vanilla perceptron algorithm on random examples from $\tilde{\mathcal{D}}$ for any number of iterations. Suppose that we run the "Update by Disagreement" algorithm for an additional arbitrary number of iterations. Then, the error of the solution is likely to be $\Omega(\mu^3)$.*

To summarize, we see that without making additional assumptions on the data distribution, it is impossible to prove convergence of our algorithm to a good solution. In the next section we show that for natural data distributions, our algorithm converges to a very good solution.

## 5 EXPERIMENTS

We now demonstrate the merit of our suggested meta-algorithm using empirical evaluation. Our main experiment is using our algorithm with deep networks in a real-world scenario of noisy labels.

In particular, we use a hypothesis class of deep networks and a Stochastic Gradient Descent with momentum as the basis update rule. The task is classifying face images according to gender. As training data, we use the Labeled Faces in the Wild (LFW) dataset for which we had a labeling of the name of the face, but we did not have gender labeling. To construct gender labels, we used an external service that provides gender labels based on names. This process resulted in noisy labels. We show that our method leads to state-of-the-art results on this task, compared to competing noise robustness methods. We also performed controlled experiments to demonstrate our algorithm's performance on linear classification with varying levels of noise. These results are detailed in Appendix B.

## 5.1 Deep Learning

We have applied our algorithm with a Stochastic Gradient Descent (SGD) with momentum as the base update rule on the task of labeling images of faces according to gender. The images were taken from the Labeled Faces in the Wild (LFW) benchmark [18]. This benchmark consists of 13,233 images of 5,749 different people collected from the web, labeled with the name of the person in the picture. Since the gender of each subject is not provided, we follow the method of [25] and use a service that determines a person's gender by their name (if it is recognized), along with a confidence level. This method gives rise to "natural" noisy labels due to "unisex" names, and therefore allows us to experiment with a real-world setup of dataset with noisy labels.

| Name | Kim | Morgan | Joan | Leslie |
|---|---|---|---|---|
| Confidence | 88% | 64% | 82% | 88% |
| Correct | 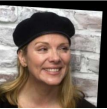 |  |  |  |
| Mislabeled |  |  |  |  |

Figure 1: Images from the dataset tagged as female

We have constructed train and test sets as follows. We first took all the individuals on which the gender service gave 100% confidence. We divided this set at random into three subsets of equal size, denoted $N_1, N_2, N_3$. We denote by $N_4$ the individuals on which the confidence level is in $[90\%, 100\%)$, and by $N_5$ the individuals on which the confidence level is in $[0\%, 90\%)$. Needless to say that all the sets $N_1, \ldots, N_5$ have zero intersection with each other.

We repeated each experiment three times, where in every time we used a different $N_i$ as the test set, for $i \in \{1, 2, 3\}$. Suppose $N_1$ is the test set, then for the training set we used two configurations:

1. A dataset consisting of all the images that belong to names in $N_2, N_3, N_4, N_5$, where unrecognized names were labeled as male (since the majority of subjects in LFW are males).

2. A dataset consisting of all the images that belong to names in $N_2, N_3, N_4$.

We use a network architecture suggested by [24], using an available tensorflow implementation[1]. It should be noted that we did not change any parameters of the network architecture or the optimization process, and use the default parameters in the implementation. Since the amount of male and female subjects in the dataset is not balanced, we use an objective of maximizing the balanced accuracy [9] - the average accuracy obtained on either class.

Training is done for 30,000 iterations on 128 examples mini-batch. In order to make the networks disagreement meaningful, we initialize the two networks by training both of them normally (updating on all the examples) until iteration #5000, where we switch to training with the "Update by Disagreement" rule. Due to the fact that we are not updating on all examples, we decrease the weight of batches with less than 10% of the original examples in the original batch to stabilize gradients. [2].

We inspect the balanced accuracy on our test data during the training process, comparing our method to a vanilla neural network training, as well as to soft and hard bootstrapping described in [33] and to the s-model described in [15], all of which are using the same network architecture. We use the initialization parameters for [33, 15] that were suggested in the original papers. We show that while in other methods, the accuracy effectively decreases during the training process due to overfitting the noisy labels, in our method this effect is less substantial, allowing the network to keep improving.

We study two different scenarios, one in which a small clean test data is available for model selection, and therefore we can choose the iteration with best test accuracy, and a more realistic scenario where there is no clean test data at hand. For the first scenario, we observe the balanced accuracy of the best available iteration. For the second scenario, we observe the balanced accuracy of the last iteration.

As can be seen in Figure 2 and the supplementary results listed in Table 1 in Appendix B, our method outperforms the other methods in both situations. This is true for both datasets, although, as expected, the improvement in performance is less substantial on the cleaner dataset.

The second best algorithm is the s-model described in [15]. Since our method can be applied to any base algorithm, we also applied our method on top of the s-model. This yields even better performance, especially when the data is less noisy, where we obtain a significant improvement.

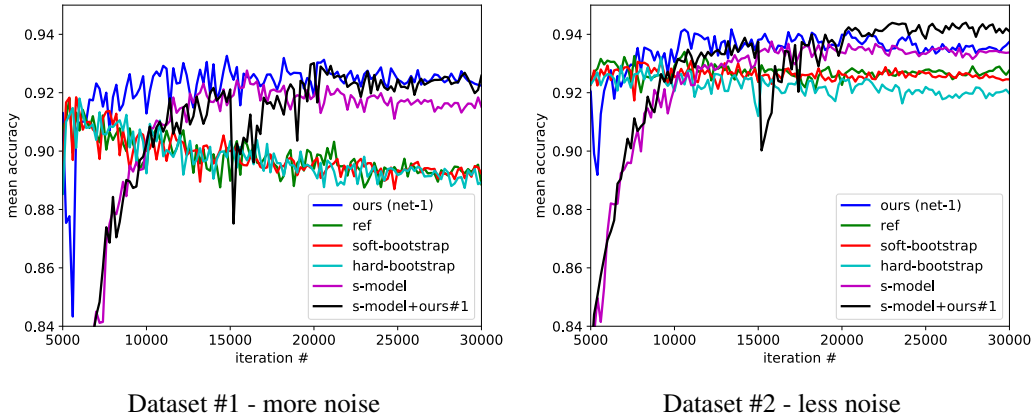

Dataset #1 - more noise          Dataset #2 - less noise

Figure 2: Balanced accuracy of all methods on clean test data, trained on the two different datasets.

# 6   Discussion

We have described an extremely simple approach for supervised learning in the presence of noisy labels. The basic idea is to decouple the "when to update" rule from the "how to update" rule. We achieve this by maintaining two predictors, and update based on their disagreement. We have shown that this simple approach leads to state-of-the-art results.

Our theoretical analysis shows that the approach leads to fast convergence rate when the underlying update rule is the perceptron. We have also shown that proving that the method converges to an optimal solution must rely on distributional assumptions. There are several immediate open questions that we leave to future work. First, suggesting distributional assumptions that are likely to hold in practice and proving that the algorithm converges to an optimal solution under these assumptions. Second, extending the convergence proof beyond linear predictors. While obtaining absolute convergence guarantees seems beyond reach at the moment, coming up with oracle based convergence guarantees may be feasible.

**Acknowledgements:**   This research is supported by the European Research Council (TheoryDL project).

## Footnotes

[1]https://github.com/dpressel/rude-carnie.

[2]Code is available online on https://github.com/emalach/UpdateByDisagreement.

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
