[Supplementary Material · labelnoise_nips supplementary.pdf]

# Supplementary material for the paper "Decoupling when to update from how to update"

## A Proofs

**Proof** of Lemma 1:

Let the distribution over instances be concentrated uniformly over the vectors of the standard basis, $e_1, \ldots, e_d$. Let $w^*$ be any vector in $\{\pm 1\}^d$. Fix some $i$. Then, with probability $1/4$ over the choice of $w_0^{(1)}, w_0^{(2)}$, we have that the signs of $\langle w_0^{(1)}, e_i \rangle, \langle w_0^{(2)}, e_i \rangle$ agree with each other, but disagree with $\langle w^*, e_i \rangle$. It is easy to see that the $i$'th coordinate of $w^{(1)}$ and $w^{(2)}$ will never be updated. Therefore, no matter how many iterations we will perform, the solution will be wrong on $e_i$. It follows that the probability of error is lower bounded by the random variable $\frac{1}{d} \sum_{i=1}^d Z_i$, the $Z_i$ are i.i.d. Bernoulli variables with $\mathbb{P}[Z_i = 1] = 1/4$. Using Chernoff's inequality,

$$\mathbb{P}\left[\frac{1}{d}\sum_{i=1}^d Z_i < 1/8\right] \leq \exp(-dC),$$

where $C = \frac{3}{112}$. It follows that if $d \geq \log(1/\delta)/C$ then with probability of at least $1 - \delta$ we will have that the error of the solution is at least $1/8$. ∎

**Proof** of Lemma 2:

Let $w_t$ be a random vector indicating the vector of the perceptron after $t$ iterations. Fix some $i$ and w.l.o.g. assume that $w_i^* = 1$. The value of $w_t$ at the $i$'th coordinate is always in the set $\{-1, 0, 1\}$. Furthermore, it alters its value like a Markov chain with a transition matrix of

$$P = \begin{pmatrix} \mu & 1-\mu & 0 \\ \mu & 0 & 1-\mu \\ 0 & \mu & 1-\mu \end{pmatrix}$$

It is easy to verify that the stationary distribution over $\{-1, 0, 1\}$ is

$$\pi = \left( \frac{\mu^2}{\mu + (1-\mu)^2}, \frac{\mu(1-\mu)}{\mu + (1-\mu)^2}, \frac{(1-\mu)^2}{\mu + (1-\mu)^2} \right).$$

Now, the probability that our algorithm will fail on the $i$'th coordinate is lower bounded by the probability that the $i$'th coordinate of both $w^{(1)}, w^{(2)}$ will be 0 and then our algorithm will see a flipped label. This would happen with probability of order of $\mu^3$ for a small $\mu$. ∎

# B    Experimental Results

We show our algorithm's performance in two controlled setups, using a perceptron based algorithm. In the first setup we test we run our algorithm on synthetic data that is generated by randomly sampling instances from the unit ball in $\mathbb{R}^d$, with different probabilities for random label-flips. In the second setup we test our performance on a binary classification task based on the MNIST dataset, again with random label-flips with different probabilities. We show that in both scenarios, our adaptation of the perceptron algorithm results in resilience for large noise probabilities, unlike the vanilla perceptron algorithm which fails to converge on even small amounts of noise.

## B.1    Linear Classification on Synthetic Data

To test the performance of the suggested perceptron-like algorithm, we use synthetic data in various dimensions, generated in the following process:

1. Randomly choose $w^* \in \mathbb{R}^d$ with a given norm $\|w^*\| = 10^3$

    (a) In each iteration, draw vectors $x \in \mathbb{R}^d$ from the uniform distribution on the unit ball until $|\langle w^*, x \rangle| \geq 1$, and then set $y = \text{sign}(\langle w^*, x \rangle)$.
    (b) With probability $\mu < 0.5$, flip the sign of $y$.

The above was performed for different values of $\mu$, and repeated 5 times for each setup. In Figure 3 we depict the average performance over the 5 runs. As can be seen, our algorithm greatly improves the noise resilience of the vanilla perceptron.

$$d = 100, \ \mu = 0.01 \qquad\qquad d = 100, \ \mu = 0.4$$

Figure 3: Mean accuracy of our algorithm (blue line) compared to a vanilla perceptron update rule (green line), averaged across 5 randomly initialized training sessions, testing different noise rate values. Each iteration is tested against a test set of 10K correctly labeled examples.

## B.2 Linear Classification on MNIST Data Noisy Labels

Here we use a binary classification task of discriminating between the digits 4 and 7, from the MNIST dataset.

We tested the performance of the above algorithm against the regular perceptron algorithm with various levels of noise.

$\mu = 0.1$

$\mu = 0.2$

$\mu = 0.3$

$\mu = 0.4$

| $\mu =$ | 0.1 | 0.2 | 0.3 | 0.4 |
|---|---|---|---|---|
| best acc. (ours) | **99.4** | **99.2** | **99.0** | **98.7** |
| best acc. (perceptron) | 97.2 | 95.0 | 91.8 | 85.8 |
| mean last 100 iters (ours) | **99.3** | **99.1** | **98.9** | **98.4** |
| mean last 100 iters (perc.) | 87.0 | 77.7 | 65.4 | 59.2 |

Figure 4: Mean accuracy of our algorithm (blue line) compared to a regular perceptron update rule (green line), with different noise rates. In all training sessions we performed 1M iterations, randomly drawing examples from the MNIST train set, and testing the accuracy of both algorithms on the MNIST test set every 1000 iterations.

## B.3   Deep Learning Detailed Results

The table below details the results of the LFW experiment, showing the balanced accuracy of all the different methods for dealing with noisy labels. We show the results on the best iteration and on the last iteration. We observe that our method outperforms other alternative, and combining it with the s-model of [15] results in an even better improvement.

Table 1: Accuracy on the test data in the best iteration (with respect to the test data) and the last iteration, achieved by each method during the training process.

| Dataset #1 | Accuracy (best iteration) | | | Accuracy (last iteration) | | |
|---|---|---|---|---|---|---|
| | Male | Female | **Mean** | Male | Female | **Mean** |
| ours (net #1) | $94.4 \pm 0.7$ | $92.7 \pm 0.2$ | $\mathbf{93.6 \pm 0.2}$ | $94.8 \pm 0.8$ | $89.7 \pm 1.3$ | $\mathbf{92.2 \pm 0.6}$ |
| ours (net #2) | $93.5 \pm 1.1$ | $93.2 \pm 0.6$ | $93.4 \pm 0.3$ | $93.7 \pm 0.8$ | $90.1 \pm 0.9$ | $91.9 \pm 0.4$ |
| s-model+ours #1 | $93.3 \pm 1.7$ | $93.8 \pm 1.4$ | $\mathbf{93.6 \pm 0.4}$ | $93.7 \pm 1.1$ | $91.4 \pm 1.0$ | $\mathbf{92.6 \pm 0.1}$ |
| s-model+ours #2 | $94.2 \pm 0.7$ | $91.7 \pm 0.6$ | $93.0 \pm 0.2$ | $93.6 \pm 1.3$ | $91.6 \pm 1.5$ | $92.6 \pm 0.1$ |
| baseline | $91.6 \pm 2.2$ | $92.7 \pm 1.8$ | $92.2 \pm 0.2$ | $94.5 \pm 0.7$ | $83.3 \pm 3.2$ | $88.9 \pm 1.3$ |
| bootstrap-soft | $92.5 \pm 0.6$ | $91.9 \pm 0.6$ | $92.2 \pm 0.2$ | $94.5 \pm 0.7$ | $84.0 \pm 1.7$ | $89.2 \pm 0.8$ |
| bootstrap-hard | $92.4 \pm 0.7$ | $91.9 \pm 1.0$ | $92.1 \pm 0.3$ | $94.7 \pm 0.2$ | $83.2 \pm 1.7$ | $88.9 \pm 0.8$ |
| s-model | $94.5 \pm 0.7$ | $91.3 \pm 0.4$ | $92.9 \pm 0.5$ | $93.3 \pm 2.0$ | $89.8 \pm 1.3$ | $91.5 \pm 0.4$ |

| Dataset #2 | Accuracy (best iteration) | | | Accuracy (last iteration) | | |
|---|---|---|---|---|---|---|
| | Male | Female | **Mean** | Male | Female | **Mean** |
| ours (net #1) | $95.5 \pm 0.8$ | $93.6 \pm 0.9$ | $94.5 \pm 0.2$ | $95.4 \pm 1.1$ | $92.1 \pm 0.7$ | $93.7 \pm 0.2$ |
| ours (net #2) | $95.7 \pm 1.5$ | $93.0 \pm 1.8$ | $94.4 \pm 0.2$ | $95.9 \pm 0.6$ | $91.6 \pm 0.6$ | $93.7 \pm 0.3$ |
| s-model+ours #1 | $95.5 \pm 0.5$ | $94.0 \pm 0.7$ | $\mathbf{94.8 \pm 0.2}$ | $95.3 \pm 1.3$ | $92.9 \pm 2.2$ | $\mathbf{94.1 \pm 0.4}$ |
| s-model+ours #2 | $95.1 \pm 0.8$ | $93.9 \pm 1.5$ | $94.5 \pm 0.3$ | $95.6 \pm 1.2$ | $92.5 \pm 1.7$ | $94.0 \pm 0.2$ |
| baseline | $93.6 \pm 0.7$ | $93.9 \pm 0.8$ | $93.8 \pm 0.3$ | $96.2 \pm 0.2$ | $89.4 \pm 1.6$ | $92.8 \pm 0.8$ |
| bootstrap-soft | $94.8 \pm 1.0$ | $92.2 \pm 0.6$ | $93.5 \pm 0.4$ | $96.2 \pm 0.6$ | $88.7 \pm 2.0$ | $92.5 \pm 0.7$ |
| bootstrap-hard | $93.9 \pm 1.2$ | $92.8 \pm 0.7$ | $93.4 \pm 0.4$ | $96.1 \pm 0.3$ | $87.9 \pm 1.6$ | $92.0 \pm 0.6$ |
| s-model | $94.8 \pm 1.0$ | $93.3 \pm 0.4$ | $94.1 \pm 0.3$ | $94.5 \pm 0.6$ | $92.3 \pm 0.2$ | $93.4 \pm 0.4$ |