[Reviews · NeurIPS 2017]

Reviewer 1



This paper proposed to train a classifier such as DNN using noising training data. The key idea is to maintain two predictors, and update based on the two predictors' disagreement. + Pros Theoretical analysis shows under mild conditions, the proposed method will converge - Cons - The overall idea is similar to the semi-supervised learning method co-training, which the authors cited but no detailed discussion. Although co-training is designed for semi-supervised learning, it's simple to adapt it for dealing with noisy data - The experiment part is very weak. Only an example on very small gender classification data will not lead to a claim "this simple approach leads to state-of-the- art results." Though theoretical analysis shows it converges, lack of experiment results makes it difficult to see how useful this method is

Reviewer 2



The authors propose a relatively simple approach to dealing with noisy labels in deep learning settings by only updating on samples for which two classifiers disagree. A theoretical foundation is shown in the case of linearly separable data. The authors then empirically validate their method on a complex face dataset. Results show improvement over baseline methods and show that they can be combined with baseline methods for further improvement. Comments - The literature review does a good job of placing the idea in context - Method appears general, scalable, and is demonstrated to be effective - Paper demonstrates that the algorithm can converge - Some of the theoretical results, even in the linear case, don't justify the method without assumptions on the data distribution. The foundations laid out however can form the bases of future theoretical analysis of this algorithm.

Reviewer 3



Summary The paper proposes a meta algorithm for training any binary classifier in a manner that is robust to label noise. A model trained with noisy labels will overfit them trained for too long. Instead, one can train two models at the same time, initialized at random, and update by disagreement: the updates are performed only when the two models' prediction differ, a sign that they are still learning from the genuine signal in the data (not the noise); and instead, defensively, if the models agree on their predictions and the respective ground truth label is different, they should not perform an update, because this is a sign of potential label noise. A key element is the random initialization of the models, since the assumption is that the two should not give the same prediction unless they are close to converge; this fits well with deep neural networks, the target of this work. The paper provides a proof of convergence in the case of linear models (updated with perceptron algorithm and in the realizable case) and a proof that the optimal model cannot be reach in general, unless we resort to restrictive distributional assumptions (this is nice since it also shows a theoretical limitation of the meta-algorithm). The method works well in practice in avoiding to overfit labels noise, as shown by experiments with deep neural networks applied to gender classification on the LFW dataset, and additional tests with linear models and on MNIST. Comments The paper is very well written: the method is introduced intuitively, posed in the context of the relevant literature, proven to be sound in a simple theoretical setting and shown to be effective on a simple experimental set up, in realistic scenario with noise. Additionally, this work stands out from the large set of papers on the topic because of its simplicity and the potential of use in conjunction with others methods. Proofs are easy to follow and seem flawless. Experimental results are promising on simple scenarios but will need future work to investigate the robustness and effectiveness on at scale and in multi-class --- although I don't consider this a major issue because the paper is well balanced between theory and practice. Also, do you find any interesting relation with boosting algorithms? In particular I am referring to "The strenght of weak learnability" by R. Schapire, that introduced a first form (although impractical) of boosting of weak classifier. The algorithm presented back then uses a form of "update by disagreement" for boosting, essentially training a third model only on data points classified the same way by the former two models. Minors 25 -> better 107 -> probabilistic 183 -> algorithm